# Contrastive Clustering to Mine Pseudo Parallel Data for Unsupervised Translation

**Xuan-Phi Nguyen**[‡,*] **Hongyu Gong**[†], **Yun Tang**[†], **Changhan Wang**[†], **Philipp Koehn**[§,†]
**& Shafiq Joty**[‡]
[†]Meta AI
[‡]Nanyang Technological University
[§]Johns Hopkins University
nguyenxu002@e.ntu.edu.sg

## Abstract

Modern unsupervised machine translation systems mostly train their models by generating synthetic parallel training data from large unlabeled monolingual corpora of different languages through various means, such as iterative back-translation. However, there may exist small amount of actual parallel data hidden in the sea of unlabeled data, which has not been exploited. We develop a new fine-tuning objective, called Language-Agnostic Constraint for SwAV loss, or LAgSwAV, which enables a pre-trained model to extract such pseudo-parallel data from the monolingual corpora in a fully unsupervised manner. We then propose an effective strategy to utilize the obtained synthetic data to augment unsupervised machine translation. Our method achieves the state of the art in the WMT'14 English-French, WMT'16 German-English and English-Romanian bilingual unsupervised translation tasks, with 40.2, 36.8, and 37.0 BLEU, respectively. We also achieve substantial improvements in the FLoRes low-resource English-Nepali and English-Sinhala unsupervised tasks with 5.3 and 5.4 BLEU, respectively.

## 1 Introduction

The quest to build a fully unsupervised machine translation (UMT) system, where only unlabeled monolingual corpora are available, has received increasing attention in recent years. Profoundly, Lample et al. (2018a;c) introduced the general principles for UMT that include cross-lingual initialization, language modeling and iterative back-translation. Although various UMT variants (Lample et al., 2018a;c; Conneau & Lample, 2019; Song et al., 2019; Liu et al., 2020; Nguyen et al., 2021) applied these principles differently, they ultimately train their models on noisy synthetic parallel data generated by the models themselves or some randomization processes, which may cause harm as the generated synthetic data is often noisy and low-quality, especially at the beginning of training. However, these methods may have missed out a potential that some parallel (or comparable) data may exist in the monolingual corpora, which can be effectively mined to augment the UMT training.

This paper is motivated to explore mining of pseudo-parallel data for UMT tasks. While there have been limited research in unsupervised mining of such data (Wu et al., 2019a;b), there have been several studies on bitext mining from sentence embeddings in semi-supervised translation or zero-shot transfer setups (Zweigenbaum et al., 2018; Schwenk, 2018; Artetxe & Schwenk, 2019a;b). However, these methods require the models to be pre-trained on massive multilingual **parallel** data, which renders them inapplicable and incomparable to the fully unsupervised setup. Furthermore, such models may not behave well off-the-shell on low-resource languages that were not in the pre-training data, such as Nepali and Sinhala (Guzmán et al., 2019). Meanwhile, applying these mining algorithms directly to sentence embeddings produced by fully self-supervised models (Conneau & Lample, 2019) leads to a significant amount of misalignments, which hurts the UMT models as shown later in our experiments (§4).

---

[*]Most of work done during an internship at Meta AI.

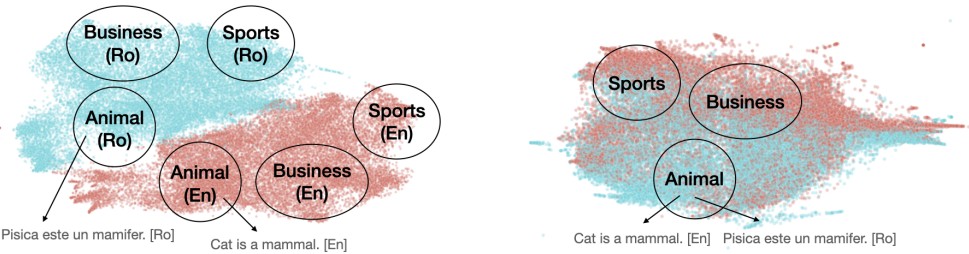

(a) Contrastive but not language-agnostic  (b) Contrastive and language-agnostic

Figure 1: Illustrations of semantically contrastive and non-language-agnostic (1a) vs. language-agnostic (1b) representations of English (red) and Romanian (cyan) sentences. For semantically contrastive models, sentences of the same topic will be clustered in the same region. If the models are also language-agnostic, the semantic regions will not differentiate sentences by language.

In this paper, we propose a novel modification to SwAV loss (Caron et al., 2020), called *Language-Agnostic Constraint for SwAV* (*LAgSwAV*)[1], which is used to fine-tune existing pre-trained self-supervised models (Conneau & Lample, 2019; Liu et al., 2020), so that they become *semantically contrastive* and *language-agnostic*. Specifically, semantic contrastiveness is defined, in this paper, as the model's ability to cluster sentences of similar semantic contents into similar groups. Meanwhile, language-agnostic property means that the embedding space of sentences from different languages are indistinguishably mixed together. Figure 1 further illustrates the two properties. We argue that these two properties combined can help the model mine more accurate pseudo-parallel data, since language-agnosticism reduces the distances between cross-lingual embeddings while semantic contrastiveness helps bring sentences of similar contents closer together **even if they are in different languages**. To achieve the semantically contrastive property, we adopt the SwAV loss (Caron et al., 2020), which is designed to classify input data into different clusters that reflect certain semantic characteristics. However, since SwAV is not designed to be language-agnostic, we introduce a new language-agnostic constraint for it to enable the latter property. Analysis experiments in §4 show that our constraint significantly outperforms vanilla SwAV loss as well as other baselines in mining pseudo-parallel data from monolingual corpora.

After fine-tuning pre-trained self-supervised models with LAgSwAV loss, we sample sentences from both languages and pass them to the model to obtain the SwAV cluster assignment probabilities, which are treated as sentence embeddings. The soft cluster assignments are then matched by a margin-based algorithm (Artetxe & Schwenk, 2019b) with additional filtering criteria (§3.2). To augment UMT training with the obtained pseudo-parallel data, we propose a simple ranking-based weighted cross-entropy loss to train on this data along with the standard iterative back-translation (§3.3). We also adopt a dynamic weight to the augmentation loss to alleviate the overfitting effect due to the limited augmentation data.

We tested our method on the standard bilingual WMT'14 English-French, WMT'16 German-English and English-Romanian unsupervised translation tasks and achieved the state of the art with 40.2, 36.8, and 37.0 BLEU, respectively; which is up to 2 BLEU gain from the baselines. In the FLo-Res low-resource unsupervised setups (Guzmán et al., 2019), our method also outperforms mBART baseline (Liu et al., 2020) by up to 2.8 BLEU in the English-Nepali and English-Sinhala tasks. We conducted a series of analyses to demonstrate the outperforming ability of LAgSwAV in mining pseudo-parallel data. The method is also found to outperform supervised LASER (Artetxe & Schwenk, 2019a), which can be attributed to LASER's lack of language specifiers.

## 2 BACKGROUND

### 2.1 UNSUPERVISED MACHINE TRANSLATION

Lample et al. (2018a) and Artetxe et al. (2018) were among the first to employ iterative back-translation in fully unsupervised machine translation (UMT). Lample et al. (2018c) later formulated the three-principles to UMT: initialization, language modeling and iterative back-translation (Sennrich et al., 2016a). Initialization can takes form of MUSE bilingual dictionary (Lample et al.,

---

[1]Code: https://github.com/nxphi47/fairseq/tree/swav_umt

2018b;a), cross-lingual masked language model (XLM) (Conneau & Lample, 2019) or Seq2Seq model (Song et al., 2019). Language modeling often takes form of denoising autoencoding (Lample et al., 2018a) or is omitted entirely (Song et al., 2019). Apart from these principles, Nguyen et al. (2021) use two distinct UMT teachers to distill the student in a two-stage back-translation process. Despite being different, most existing methods commonly employ the same iterative back-translation process. Multilinguality is also proved to be useful as Liu et al. (2020) and Garcia et al. (2021) utilize extra training data of more than two languages. These methods, however, do not leverage the potential pseudo-parallel data that may exist in the monolingual data to improve UMT.

## 2.2 PSEUDO-PARALLEL DATA MINING

Parallel data mining for machine translation has been an active field of research (Zweigenbaum et al., 2018; Schwenk, 2018; Chinea-Ríos et al., 2017). Artetxe & Schwenk (2019b) suggested an effective margin-based algorithm to mine parallel data using sentence embeddings, which we also use in our method. More recently, LASER (Artetxe & Schwenk, 2019a) was trained on massive multilingual parallel data from 93 languages. Nonetheless, most of the effort has been invested largely in a supervised setup where the training data is parallel. In terms of UMT, Wu et al. (2019a) proposed an *extract-edit* approach to mine relevant data and edit it for use, while Wu et al. (2019b) mine weakly-paired documents. Tran et al. (2020) suggested to train on massive multilingual data from 25 languages and mine data from 180 translation directions, which may not be applicable in bilingual setup, where monolingual data only for the two languages under consideration are available.

## 2.3 SWAPPING ASSIGNMENTS BETWEEN VIEWS (SWAV LOSS)

Caron et al. (2020) introduced Swapping Assignments between Views (SwAV) loss as a means to conduct self-supervised training in computer vision because masking-based training methods applied in language modeling are intractable in images. Although SwAV is a type of contrastive learning, it differs from its sisters (Chen et al., 2020; He et al., 2020; Wu et al., 2018) in that it is an online clustering-based method. This means the model is trained to assign cluster labels, or "code" vector $q$, to an input image with encoded latent features $z$. In particular, SwAV loss seeks to enforce consistency between codes of different augmentations (or views) of the same image. In other words, different augmentations of the same image should have almost the same semantic content as the original, and thus the cluster assignments of them should also be consistently the same.

Formally, let $z_1, z_2 \in \mathbb{R}^d$ be latent features of two distinct views $X_1, X_2$ of the same image $X$ after passing them through an encoder $f_\theta$, we compute the corresponding codes $q_1, q_2 \in [0, 1]^K$ by passing $z_1, z_2$ into the prototype layer, which consists of prototype weights $C = [c_1, ..., c_K]$ with $c_i \in \mathbb{R}^d$. We then proceed to *swap* the codes $q_1, q_2$ and use them as labels for $z_2, z_1$, respectively:

$$\mathcal{L}_{\text{SwAV}}(z_1, z_2) = l(z_1, q_2) + l(z_2, q_1) \tag{1}$$

The loss function $l(z, q)$ in Equation (1) measures the similarity distance between features $z$ and code $q$, and is defined as:

$$l(z, q) = -\sum_k q^k \log p^k, \ \text{with} \ p^k = \frac{\exp(\frac{1}{\tau} z^T c_k)}{\sum_{k'} \exp(\frac{1}{\tau} z^T c_{k'})} \tag{2}$$

where $\tau$ is a temperature hyper-parameter and $k$ is the index for $c_k$ row in $C$. Figure 2a illustrates how SwAV loss works. While the above formulations involve two different augmentations $X_1, X_2$ of the same image for brevity purpose, multiple augmentations are used in practice. We refer the reader to (Caron et al., 2020) for further details on SwAV loss.

## 3 METHOD

In this section, we present our three main contributions: the language-agnostic constraint for SwAV loss or LAgSwAV (§3.1), the pseudo-parallel data mining with filter suite (§3.2), and finally the rank-based cross-entropy loss for UMT training with augmented data (§3.3).

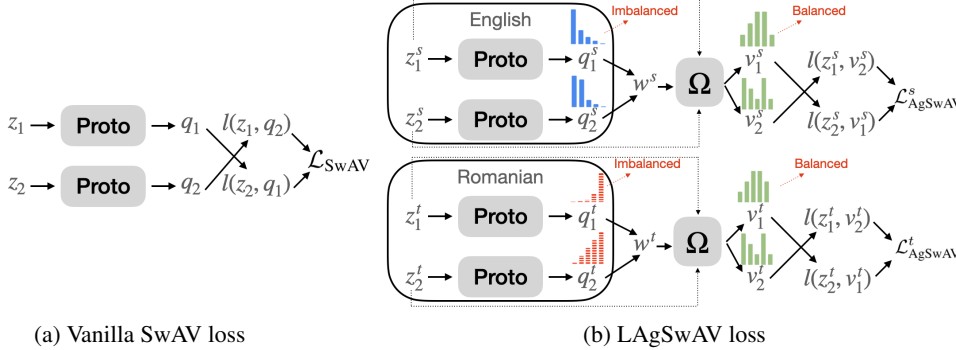

(a) Vanilla SwAV loss             (b) LAgSwAV loss

Figure 2: Comparison between vanilla SwAV (2a) and LAgSwAV (2b) losses. The "Proto" block represents the prototype layer $C$. Unlike SwAV, our LAgSwAV loss differentiates which language a sentence belongs to. As in Figure 2b, the cluster assignments of English sentences (*i.e.*, $q_1^s, q_2^s$) are skewed towards left clusters (blue histogram), and those of Romanian (*i.e.*, $q_1^t, q_2^t$) are skewed towards right clusters (red dotted histogram), causing an imbalance. The $\Omega$ block rebalances them into more evenly distributed assignments $v_1^s, v_2^s, v_1^t$ and $v_2^t$.

### 3.1 LANGUAGE-AGNOSTIC CONSTRAINT FOR SWAV LOSS IN LANGUAGE MODEL

The SwAV loss was developed to conduct self-supervised training in vision due to infeasibility of applying masked language modeling (MLM) to images (Caron et al., 2020). Thus, for language model pre-training, we stick to MLM and use our proposed LAgSwAV to fine-tune a pre-trained LM to achieve semantic contrastiveness and language agnosticism.

Specifically, in the presence of sentences from two languages, let $\mathcal{B}^s = [X_1, ..., X_B]$ and $\mathcal{B}^t = [Y_1, ..., Y_B]$ be the batches of $B$ sentences from language $s$ and $t$, respectively. We proceed to augment them into $\mathcal{B}^{s,1} = [X_1^1, ..., X_B^1]$ and $\mathcal{B}^{s,2} = [X_1^2, ..., X_B^2]$, and $\mathcal{B}^{t,1} = [Y_1^1, ..., Y_B^1]$ and $\mathcal{B}^{t,2} = [Y_1^2, ..., Y_B^2]$ by various noising techniques, such as token swapping, masking and deleting. Next, we denote $\mathcal{F}_\theta$ as the Transformer encoder (Conneau & Lample, 2019), which outputs the sentence embedding $z$ given an input sentence. We compute the respective latent features $Z^{s,1}, Z^{s,2}, Z^{t,1}, Z^{t,2} \in \mathbb{R}^{B \times d}$ for $\mathcal{B}^{s,1}, \mathcal{B}^{s,2}, \mathcal{B}^{t,1}, \mathcal{B}^{t,2}$ using $\mathcal{F}_\theta$, as described in §2.3. Subsequently, we compute the code cluster assignments $Q^{s,1}, Q^{s,2}, Q^{t,1}, Q^{t,2} \in [0,1]^{B \times K}$ as:

$$[Q^{s,1}; Q^{s,2}; Q^{t,1}; Q^{t,2}] = \text{sinkhorn}(\exp(\frac{[Z^{s,1}; Z^{s,2}; Z^{t,1}; Z^{t,2}]C^T}{\epsilon})) \tag{3}$$

where $[\bullet; \bullet]$ is the row-wise concatenation function and $\text{sinkhorn}(\bullet)$ is the iterative Sinkhorn-Knopp algorithm (Cuturi, 2013), a batch-wise renormalization process to compute the cluster assignment probabilities, and $\epsilon$ is a hyper-parameter to control the smoothness of the distribution.

The above formulation is in fact indifferent from the vanilla SwAV loss (§2.3). However, intuitively, if the model $\mathcal{F}_\theta$ is **not** language-agnostic (*i.e.*, $s$ and $t$ are distributionally separate), the resulting cluster assignments $Q^{s,j}$ and $Q^{t,j}$ are likely to be distributed unevenly for $s$ and $t$. In other words, some clusters $c_i \subset C$ will be frequently assigned to the sentences of language $s$ but rarely to the sentences of $t$, and vice versa. Figure 2b illustrates the imbalance phenomenon. We argue that it is possible to enforce the language-agnostic property indirectly by fixing this imbalance phenomenon. Therefore, the objective of our proposed **LAgSwAV** is to measure the stochastic imbalance in cluster assignments $Q^{s,j}, Q^{t,j}$, and rebalance them into $V^{s,j}, V^{t,j} \in [0,1]^{B \times K}$ such that clusters in $V^{s,j}, V^{t,j}$ are stochastically assigned evenly regardless of the language of the sentences. Then, we compute the total LAgSwAV loss as usual for both $s$ and $t$ as:

$$\mathcal{L}_{\text{AgSwAV},i}^s = l(z_i^{s,1}, v_i^{s,2}) + l(z_i^{s,2}, v_i^{s,1}) \; ; \; \mathcal{L}_{\text{AgSwAV},i}^t = l(z_i^{t,1}, v_i^{t,2}) + l(z_i^{t,2}, v_i^{t,1}) \tag{4}$$

$$\mathcal{L}_{\text{LAgSwAV}} = \sum_i \mathcal{L}_{\text{AgSwAV},i}^s + \sum_i \mathcal{L}_{\text{AgSwAV},i}^t \tag{5}$$

We formalize the transformation from $Q$ to $V$ using the *language-agnostic rebalancing function* $\Omega$, which operates at the batch level to capture the statistics on cluster assignments. Specifically, given the $Q$ matrices specified above, we compute the per-language rebalance weight vectors $w^s, w^t \in \mathbb{R}^K$, such that when multiplying them with prototype outputs, they try to suppress the

magnitudes of frequently-assigned clusters and promote those of rarely-assigned clusters, resulting in more balanced cluster assignments. More formally, the weight vectors are computed as:

$$\boldsymbol{f}^s = \sum_{i=1}^{2B} \text{onehot}([\boldsymbol{Q}^{s,1}; \boldsymbol{Q}^{s,2}])_i \; ; \; \boldsymbol{f}^t = \sum_{i=1}^{2B} \text{onehot}([\boldsymbol{Q}^{t,1}; \boldsymbol{Q}^{t,2}])_i \tag{6}$$

$$\boldsymbol{w}^s = 1.0 - (\boldsymbol{f}^s / \sum_{j=1}^{K} \boldsymbol{f}_j^s) \; ; \; \boldsymbol{w}^t = 1.0 - (\boldsymbol{f}^t / \sum_{j=1}^{K} \boldsymbol{f}_j^t) \tag{7}$$

where onehot($\bullet$) is a row-wise one-hot function which returns a one-hot vector by turning the maximum entry in the row to $1.0$ and the rest to $0.0$. We then use the weight vectors $\boldsymbol{w}^s, \boldsymbol{w}^t$ to compute the rebalanced cluster assignments $\boldsymbol{V}^{s,j}, \boldsymbol{V}^{t,j} \in \mathbb{R}^{B \times K}$ by modifying Equation (3) as:

$$[\boldsymbol{V}^{s,1}; \boldsymbol{V}^{s,2}; \boldsymbol{V}^{t,1}; \boldsymbol{V}^{t,2}] = \text{sinkhorn}(\exp(\frac{1}{\epsilon}[\boldsymbol{w}^s \odot [\boldsymbol{Z}^{s,1}; \boldsymbol{Z}^{s,2}]; \boldsymbol{w}^t \odot [\boldsymbol{Z}^{t,1}; \boldsymbol{Z}^{t,2}]]\boldsymbol{C}^T)) \tag{8}$$

where $\boldsymbol{a} \odot \boldsymbol{B}$ is an element-wise multiplication operation of vector $\boldsymbol{a}$ applied to each row of $\boldsymbol{B}$, and the rows of $\boldsymbol{V}^{s,j}, \boldsymbol{V}^{t,j}$ represent the $\boldsymbol{v}_i^{s,j}, \boldsymbol{v}_i^{s,j}$ vectors in Equation (4). Note that there is no gradient flow for the computation of $\boldsymbol{w}^s$ and $\boldsymbol{w}^t$, as well as the cluster assignments $\boldsymbol{Q}$ and $\boldsymbol{V}$.

## 3.2 Pseudo-Parallel Data Mining with Filter Suite

To mine pseudo-parallel data, we sample sentences from two monolingual corpora $s$ and $t$, and compute their cluster assignments using our LAgSwAV model. The cluster assignment probabilities are treated as sentence embeddings in the next step to mine the data using the margin-based algorithm proposed by Artetxe & Schwenk (2019b). The resulting dataset is decreasingly sorted by the alignment scores. However, we empirically found that the obtained data are considerably noisy. Therefore, we propose a **filter suite**, which is a set of filtering criteria to filter out low-quality samples from the initial mined dataset, $\mathbb{X}_{s-t}^r = [(x_1, y_1), ..., (x_m, y_m)]$, resulting in the final mined data $\mathbb{X}_{s-t} = [(x_1, y_1), ..., (x_n, y_n)]$. While details and motivations for these criteria are given in the Appendix, we briefly introduce them as:

1. Minimum and maximum sentence lengths of $L_{\min}$ and $L_{\max}$ for both source and target.

2. Source-target length ratio $\mu$ with $\mu \geq 1.0$; *i.e.,* $\frac{1}{\mu} \leq \frac{\text{len}(x_i)}{\text{len}(y_i)} \leq \mu$.

3. Subword overlap ratio $\gamma_i$ between source and target is larger than the corpus mean value, where $\gamma_i$ is defined as $\frac{\text{len}(x_i \cap y_i)}{\text{len}(x_i) + \text{len}(y_i)}$; *i.e.,* $\gamma_i > \sum_j^n \frac{\gamma_j}{n}$.

4. Accept only top $\rho\%$ of pairs based on the alignment scores. This is equivalent to threshold filtering (Artetxe & Schwenk, 2019b), but it does not depend on an absolute threshold value.

5. UMT filtering with agreement BLEU $> \beta$. Specifically, after training a UMT model with the mined data from steps 1-4, we use the model itself to translate each source sentence of the mined data and measure the BLEU score between its translation and the target. We then filter out any pair with BLEU $< \beta$. This step drastically reduces the size of the mined dataset.

## 3.3 Rank-based Cross-Entropy for UMT Integration

It is trivial to compute a supervised loss on the mined pseudo-parallel data and add it to the standard UMT back-translation loss. However, we found that (*i*) the mined data still contains, especially at the bottom of the ranked list, certain amount of misalignments, and (*ii*) the data is too small ($<$100K sentence pairs) to not cause overfitting. These issues motivate us to develop a *Rank-based Cross-Entropy* loss, or *Rank-XE*, to be applied to the mined data. Specifically, given a mined dataset $\mathbb{X}_{s-t} = [(x_1, y_1), ..., (x_n, y_n)]$, which is decreasingly sorted by their alignment scores, we assign a weight $\sigma_i = 1.0 - (i-1)/n$ for each pair $(x_i, y_i)$. $\sigma_i$ is designed to assign higher weights to the pairs at the top and lower to the bottom ones, which effectively reduces the influence of low-ranked pairs to the loss function. The loss for the mined (or augmented) data is then defined as:

$$\mathcal{L}_{\text{AUG}} = \sum_i -\sigma_i \log P_\theta(y_i | x_i) \tag{9}$$

which is then added to the standard iterative back-translation loss $\mathcal{L}_{\text{BT}}$ to compute the total loss:

$$\mathcal{L} = \mathcal{L}_{\text{BT}} + \lambda_{\text{aug}} \mathcal{L}_{\text{AUG}} \tag{10}$$

where the weight $\lambda_{\text{aug}}$ is set to 1 at the beginning and gradually decreases to 0 to avoid overfitting.

Table 1: BLEU scores on the bilingual WMT'14 English-French (En-Fr), WMT'16 English-German (En-De) and WMT'16 English-Romanian (En-Ro) UMT tasks. Note that mBART is incomparable as it was trained on *multilingual* CC25 data, 45 times large than the *bilingual* NewsCrawl data used in our experiments. The numbers in subscripts are the corresponding *sacrebleu* scores.

| Method | En-Fr | Fr-En | En-De | De-En | En-Ro | Ro-En |
|---|---|---|---|---|---|---|
| NMT (Lample et al., 2018c) | 25.1 | 24.2 | 17.2 | 21.0 | 21.1 | 19.4 |
| PBSMT (Lample et al., 2018c) | 27.8 | 27.2 | 17.7 | 22.6 | 21.3 | 23.0 |
| XLM (Conneau & Lample, 2019) | 33.4 | 33.3 | 26.4 | 34.3 | 33.3 | 31.8 |
| MASS (Song et al., 2019) | $37.5_{36.4}$ | $34.9_{34.5}$ | $28.3_{28.3}$ | $35.2_{35.1}$ | $35.2_{35.3}$ | $33.1_{32.7}$ |
| CBD (Nguyen et al., 2021) | $38.2_{37.2}$ | $35.5_{35.1}$ | $\mathbf{30.1_{30.1}}$ | $36.3_{36.2}$ | $36.3_{35.4}$ | $33.8_{33.5}$ |
| mBART (Liu et al., 2020)$_{*incomparable}$ | - | - | 29.8 | 34.0 | 35.0 | 30.5 |
| CRISS (Tran et al., 2020)$_{*incomparable}$ | 38.3 | 36.3 | 32.1 | 37.1 | 35.1 | 37.6 |
| LAgSwAV (MASS) | $\mathbf{40.2}_{\,39.0}$ | $\mathbf{37.6}_{\,37.4}$ | $29.5_{\,29.7}$ | $\mathbf{36.8}_{\,36.7}$ | $\mathbf{37.0}_{\,37.1}$ | $\mathbf{34.9}_{\,34.7}$ |

## 4 EXPERIMENTS

In this section, we first report the performance of our method in comparison with the established baselines on the standard WMT and FLoRes low-resource UMT tasks. Then, we experimentally show how LAgSwAV is able to mine better pseudo-parallel data. We also compare our method with related approaches and conduct other analyses. To optimize for the most informative content within the limited page real estate, we leave some of experimental details to the Appendix.

### 4.1 WMT BILINGUAL UNSUPERVISED MACHINE TRANSLATION TASKS

**Setup.** For the WMT'14 English-French (En-Fr), WMT'16 English-German (En-De) and WMT'16 English-Romanian (En-Ro) bilingual UMT tasks, we follow the established predecessors (Lample et al., 2018c; Conneau & Lample, 2019; Song et al., 2019; Nguyen et al., 2021) to use only the monolingual data from 2007-2017 WMT News Crawl datasets of the two languages for each task. Note that no data from a third language is used. We fine-tune the pre-trained XLM (Conneau & Lample, 2019) with our proposed LAgSwAV loss and follow the process in §3.2 to mine pseudo-parallel data, which are then used to augment MASS UMT models (Song et al., 2019). Following all established previous work, we evaluate the models with tokenized BLEU *multi-bleu.perl* script (Koehn et al., 2007). In addition, we also report the corresponding *sacrebleu* scores (Post, 2018) in the subscript for our method as well as the baselines that we reproduced in our experiments.

**Results.** From the results reported in Table 1, we notice that our method outperforms MASS baseline (Song et al., 2019) by 2.7, 2.1, 1.2, 1.6, 1.8 and 1.8 BLEU for the WMT En-Fr, Fr-En, En-De, De-En, En-Ro and Ro-En, respectively. It also surpasses CBD (Nguyen et al., 2021) by up to 2 BLEU in 5 out of 6 language pairs. The results demonstrate that the pseudo-parallel data mined by our method is valuable for the UMT models.

### 4.2 LOW-RESOURCE UNSUPERVISED TRANSLATION

For the FLoRes low-resource UMT tasks (Guzmán et al., 2019), we follow the same setup as mBART (Liu et al., 2020). Specifically, we reuse their published pre-trained model to fine-tune with our LAgSwAV loss and mine pseudo-parallel data. Table 2 reports the results on Ne-En and Si-En for our method, as compared with XLM (Conneau & Lample, 2019), MASS (Song et al., 2019) and mBART (Liu et al., 2020). It can be shown that our method outperforms mBART by up to 2.8 BLEU in these UMT tasks.

Table 2: Performance on the low-resource FLoRes Nepali-English (Ne-En) and Sinhala-English (Si-En) UMT tasks.

| Method | Ne-En | En-Ne | Si-En | En-Si |
|---|---|---|---|---|
| XLM | 0.5 | 0.1 | 0.1 | 0.1 |
| MASS | 0.7 | 0.3 | 0.4 | 0.1 |
| mBART | 10.0 | 4.4 | 8.2 | 3.9 |
| LAgSwAV | 12.8 | 5.3 | 9.4 | 5.4 |

### 4.3 UNDERSTANDING LANGUAGE AGNOSTIC CONSTRAINT FOR SWAV LOSS

To understand how LAgSwAV behaves in terms of the aforementioned language agnostic and semantically contrastive properties, we conduct a series of evaluations with manual visualization as well as automatic metrics to measure the focused properties.

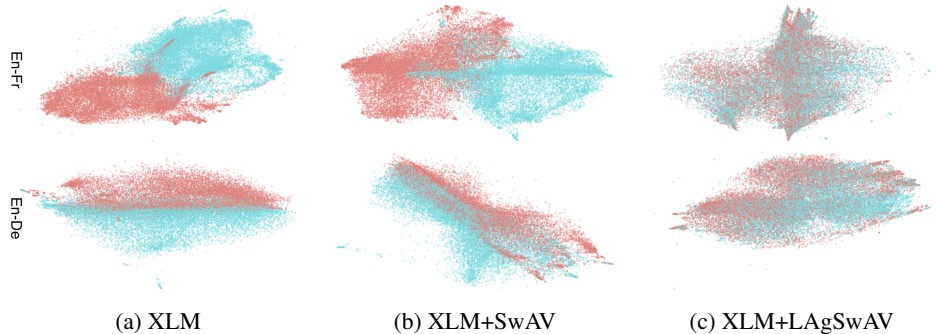

|                | (a) XLM | (b) XLM+SwAV | (c) XLM+LAgSwAV |

Figure 3: T-SNE visualizations of sentence embeddings produced by XLM, XLM+SwAV and XLM+LAgSwAV for En-Fr and En-De. Visualizations for En-Ro is given in the Appendix.

### 4.3.1 T-SNE VISUALIZATIONS

We sample 50K sentences of both languages from a held-out monolingual data for each language pair and use T-SNE (Van der Maaten & Hinton, 2008) to visualize the sentence embeddings produced by XLM, XLM fine-tuned with SwAV (XLM+SwAV) and XLM fine-tuned with LAgSwAV (XLM+LAgSwAV). From the visualizations shown in Figure 3, it can be seen that the English embeddings (red) are scattered separately and distinguishable from those of French and German (cyan) in XLM and SwAV XLM. Among them, the visualizations for German fare better than those of French. However, after fine-tuning with LAgSwAV loss, the sentence embeddings are mixed indistinguishably, indicating a high degree of language-agnosticism.

### 4.3.2 QUANTITATIVE MEASURES

Although §4.3.1 provides a qualitative sense of how the models exhibit in terms of the two examined properties, it is necessary to measure the properties in a quantifiable manner. We propose three metrics to evaluate the models: Tatoeba Accuracy, Global Accuracy and Cluster Word Distribution. We compare different models on these metrics in Table 3. In the Appendix, we also provide sentence samples of different clusters to show that they are semantically related.

**Tatoeba Accuracy.**    Artetxe & Schwenk (2019a) proposed to use parts of the Tatoeba corpus as test sets for the Tatoeba similarity search task, which can be used to measure how good a model is at mining parallel data. Specifically, for each source sentence, we use cosine similarity to search for the closest target sentence based on the sentence embeddings to check if the obtained target sentence is indeed the correct target of the given source, from which we compute the Tatoeba accuracy score.

**Global Accuracy.**  This metric uses a validation set of parallel sentences to measure how *language-agnostic* the models are, based on the perception that non-agnostic models exhibit a separation of latent spaces for different languages. Specifically, if there is a separation by language, each sentence of language $s$ would be closer to all other sentence of the same language $s$ than any sentence of language $t$. Therefore, we measure Global Accuracy in the same way as Tatoeba except that we search for the target sentence in a search space that includes not just the target sentences, but both source and target sentences combined. If the model is non-agnostic, the search will always pick incorrect intra-lingual sentences, driving down the accuracy significantly.

**Word Distribution Score.**  This metric uses a held-out monolingual data to measure how *semantically contrastive* the models are in terms of content and topics. One of the ways to measure this is to detect certain words that appear frequently in only one cluster and rarely in others. Intuitively, if a cluster contains certain key-words dominantly, it is likely the sentences in that cluster are of similar semantic topic or content. From this notion, the word distribution score is computed based on the unique word appearance frequency for each cluster. Further details are given in the Appendix.

**Results.**    Table 3 shows the results in terms of the three metrics for the XLM (baseline), XLM+SwAV, and XLM+LAgSwAV models along with their respective BLEU scores achieved using the mined data obtained by the models after going through the Filter Suite (§3.2). We observe consistent improvements across all three measures for our LAgSwAV model. In particular, the global accuracy scores reflect closely the degree of language-agnosticism as observed in Figure 3. For example, the global accuracy values for French and Romanian are particularly low for the base-

Table 3: Tatoeba, Global Accuracy (GlobAcc) and Word Distribution scores (WordDis) of baseline XLM and XLM fine-tuned with SwAV and LAgSwAV. We also include the corresponding downstream UMT performance (BLEU) of models trained with the respective augmentation data.

| Model | En-Fr | | | | En-De | | | | En-Ro | | | |
|---|---|---|---|---|---|---|---|---|---|---|---|---|
| | Tatoeba | GlobAcc | WordDis | BLEU | Tatoeba | GlobAcc | WordDis | BLEU | Tatoeba | GlobAcc | WordDis | BLEU |
| **Unsupervised models** | | | | | | | | | | | | |
| Baseline | 30.5 | 25.6 | 0.30 | 35.9 | 59.1 | 74.1 | 0.35 | 25.5 | 11.5 | 13.2 | 0.22 | 34.1 |
| SwAV | 34.2 | 30.4 | 0.28 | 36.1 | 62.8 | 76.2 | 0.34 | 26.1 | 11.9 | 24.3 | 0.18 | 34.0 |
| LAgSwAV | 91.2 | 88.2 | 0.38 | 40.2 | 93.0 | 87.2 | 0.39 | 29.5 | 80.3 | 84.6 | 0.30 | 37.0 |
| **Supervised models** | | | | | | | | | | | | |
| LASER | 96.2 | 93.8 | 0.43 | 38.3 | 99.1 | 93.9 | 0.46 | 28.2 | 98.0 | 96.8 | 0.35 | 33.3 |

Table 4: Comparison of our method LAgSwAV versus supervised LASER (Artetxe & Schwenk, 2019a), along with the study on the effect of different components of our method: Rank-based Cross Entropy (Rank-XE) and Dynamic $\lambda_{\text{aug}}$ (§3.3) and Filter Suite (§3.2).

| Method | En-Fr | Fr-En | En-De | De-En | En-Ro | Ro-En |
|---|---|---|---|---|---|---|
| Extract-Edit (Wu et al., 2019a) | 27.6 | 26.9 | 19.5 | 23.3 | 23.3 | 21.6 |
| Weakly-paird Docs (Wu et al., 2019b) | - | - | 24.2 | 30.3 | 30.1 | 27.6 |
| MASS (Song et al., 2019) | 37.5 | 34.9 | 28.3 | 35.2 | 35.2 | 33.1 |
| + LASER (Artetxe & Schwenk, 2019a) | 36.2 | 34.5 | 25.6 | 30.1 | 27.5 | 26.5 |
| + LASER + Rank-XE & $\lambda_{\text{aug}}$ | 37.4 | 35.0 | 27.0 | 33.6 | 32.9 | 32.0 |
| + LASER + Rank-XE & $\lambda_{\text{aug}}$ + Filter Suite | 38.3 | 35.7 | 28.2 | 34.1 | 36.0 | 33.3 |
| MASS | | | | | | |
| + LAgSwAV Only | 35.9 | 34.7 | 26.0 | 31.2 | 26.9 | 26.4 |
| + LAgSwAV + Rank-XE & $\lambda_{\text{aug}}$ | 37.5 | 35.0 | 28.0 | 35.0 | 35.4 | 33.0 |
| + LAgSwAV + Rank-XE & $\lambda_{\text{aug}}$ + Filter Suite | 40.2 | 37.6 | 29.5 | 36.8 | 37.0 | 34.9 |

lines, while that of German is already much higher. These improvements also translate well to the performance of our LAgSwAV models on the downstream UMT tasks.

## 4.4 COMPARISON AND ABLATION STUDY

As an unsupervised bitext mining method, it is noteworthy to compare our method with existing methods (Wu et al., 2019a;b). We also report the results for LASER (Artetxe & Schwenk, 2019a). Note that LASER is not directly comparable to our setup as it was trained on parallel data from 93 languages. Specifically, we use the pre-trained LASER model to mine pseudo-parallel data from the same monolingual sources used in LAgSwAV. We then use the mined data to augment UMT models with and without the Filter Suite (§3.2) and rank-based loss (§3.3). Plus, we also conducted an ablation study to examine the contributions of different components of our method. We train the UMT models with the mined data from our LAgSwAV models without any filtering and rank-based loss, and then subsequently add these components to see how they contribute to the performance.

**Ablation Study.** From the results in Table 4, it is noticeable that without a sufficient filtering process, the mined data by LASER as well as LAgSwAV cause significant performance drops across all language pairs. This is because the majority of sentence pairs ranked at the bottom of the mined data are noisy, causing incorrect loss terms. The performances gradually improve as we add rank-based cross entropy and dynamic $\lambda_{\text{aug}}$ weight to the framework, which essentially reduce the impact of low-ranked pairs. However, the most performance jumps are observed when we add the Filter Suite, where we considerably filter the mined data and retain the top $\approx 5\%$ of the mined data. These observations are true for both LASER and our LAgSwAV fine-tuned XLM models.

**Comparison with LASER.** A more surprising observation we have from the results in Table 4 is that LASER underperforms our method, even with the Filter Suite. This is despite the fact that LASER was trained on a much larger **parallel** data from 93 languages. After investigation, we discover that LASER **does not** differentiate languages and the monolingual data contains sentences from the other language. For instance, the English corpus contains some French sentences as artifacts; LASER simply picks these artifacts up and pair them with their *identical matches* found in the French corpus. Since these Fr-Fr pairs are almost identical, they are (wrongly) ranked on top of

Table 5: Agreement BLEU between the targets of the augmentation data mined by different pre-trained models and their translations produced by a fixed baseline MT model. Higher scores means the targets are close to the sources according to the baseline model.

| Method | En-Fr | Fr-En | En-De | De-En | En-Ro | Ro-En |
|---|---|---|---|---|---|---|
| XLM | 3.5 | 2.4 | 1.5 | 1.4 | 2.3 | 1.8 |
| XLM + SwAV | 3.4 | 1.7 | 1.9 | 1.3 | 2.5 | 2.0 |
| LASER | 24.3 | 24.7 | 13.4 | 15.6 | 17.5 | 17.6 |
| XLM + LAgSwAV | 37.9 | 36.9 | 21.2 | 24.4 | 26.3 | 27.4 |

the mined dataset. Our LAgSwAV, meanwhile, differentiates languages thanks to XLM's language embedding layer. Therefore, when it encounters a French sentence that is 'marked' as English, the model will infer it as erratic and avoid selecting it during mining. Table 5 verifies this observation.

## 4.5 FURTHER ANALYSIS

**Quality of Mined Data.** To gain further insights into the quality of the augmentation dataset mined by our method, we use a supervised NMT model (Ott et al., 2018) to translate the source side of the mined data and compute the *agreement BLEU* scores between its translations and the dataset's target side. The higher the agreement BLEU, the more likely the source and target of the mined dataset are semantically and meaningfully aligned, according to the NMT model. We use this metric to compare the quality of mined datasets produced by XLM, supervised LASER (Artetxe & Schwenk, 2019a), XLM+SwAV and XLM+LAgSwAV. As shown in Table 5, XLM and XLM+SwAV fail completely at mining pseudo-parallel data as their agreement BLEU scores are significantly low. Meanwhile, our method outperforms even supervised LASER, which aligns well with the results in Table 4.

**Percentage $\rho\%$.** Among the hyper-parameters specified in the Filter Suite (§3.2), we found that the $\rho\%$ plays the paramount role. As shown in Figure 4, for the WMT En-Ro task, we found only the top 10% of the mined data is valuable to improve UMT performance. As we gradually include more data from the bottom of the mined data by increasing $\rho$, the performance degrades substantially.

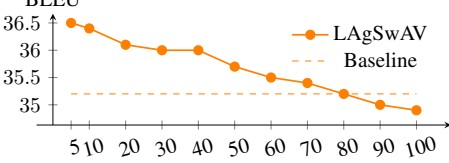

| En-De | Baseline | LAgSwAV |
|---|---|---|
| X → Y* | 24.9 | 26.3 |
| X* → Y | 25.5 | 28.5 |
| X** → Y* | 32.9 | 35.5 |

Figure 4: Performances of our method on En-Ro w.r.t filtering $\rho\%$.

Table 6: Translationese analysis on En-De unsupervised MT models

**Translationese Effect.** Edunov et al. (2020) found that back-translation only benefits translationese-source sentences but does not improve for natural-source texts. To investigate whether our method is affected by the *translationese effect*, we use the test sets provided by Edunov et al. (2020) and replicate the translationese analysis in three setups: (*i*) Natural-source → Translationese-target (X→Y*), (*ii*) Translationese-source → Natural-target (X*→Y) and (*iii*) Translationese of translationese of source → Translationese-target (X**→Y*). The results in Table 6 show that our method improves the performance in De-En and En-De UMT tasks for both natural-source and translationese-source setups. This demonstrates that our method is not affected by the translationese effect.

## 5 CONCLUSION

We have proposed a new language-agnostic SwAV loss, which is used to fine-tune pre-trained language models to improve their ability to mine pseudo-parallel data in an unsupervised manner. We have also introduced additional modules, such as Filter Suite and dynamic loss weight, to improve the performance of UMT systems. The method achieves significant improvements over the baselines in the WMT English-French, German-English and English-Romanian bilingual UMT tasks. It also outperforms mBART in the FLoRes Nepali-English and Sinhala-English UMT tasks. Our analysis also shows that the method outperforms the baselines in mining pseudo-parallel data.

## ETHICS AND REPRODUCIBILITY STATEMENTS

**Ethics.** The paper presents research on unsupervised machine translation where experiments were conducted on publicly available datasets, such as the NewsCrawl and CommonCrawl datasets, which have been used in various established publications. Thus, we believe the research and experiments do not contain any questionable concern regarding bias, discrimination, fairness, privacy, etc. Nonetheless, in case of application to production systems, we advise practitioners to use our solutions with caution, as our mining method may still align source-target pairs that are close but not exact, such as "I am from the US" versus "I am from California". These inexact pairs may then influence the downstream translation model to make incorrect predictions.

**Reproducibility.** Apart from the experimental explanations in the main paper, Appendix appendix A.3 provides further experimental details on the WMT and FLoRes unsupervised translation tasks, as well as other analysis experiments. In addition, the source codes provided above contain further technical instructions and code explanations to reproduce the results.

## ACKNOWLEDGMENTS

We deeply appreciate the efforts of our anonymous reviewers and meta-reviewer in examining and giving us feedback on our paper. We also thank Juan Pino, Paco Guzman, Naman Goyal, Chau Tran and Xian Li from Meta AI for their valuable advice and feedback for our work.

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

# A APPENDIX

In the Appendix, we provide further details and explanations about the inner-workings of our method. Plus, we also show extra experimental configurations as well as other supplementary experiments.

## A.1 FURTHER EXPLANATION FOR OUR METHOD

In this section, we first explain the Sinkhorn algorithm and more details about the Filter Suite that we use to filter pseudo-parallel data mined from our LAgSwAV model. Then, we lay out the complete procedure used to integrate the mined data with standard back-translation for unsupervised machine translation models.

### A.1.1 LANGUAGE-AGNOSTIC CONSTRAINT FOR SWAV LOSS IN LANGUAGE MODEL

In addition to the details explained in the main paper, we illustrate the batch-wise *sinkhorn* algorithm (Cuturi, 2013) in Algorithm 1. Further details about this procedure and the SwAV loss are also provided in Caron et al. (2020).

---

**Algorithm 1** Sinkhorn: Given matrix $\boldsymbol{Z} \in \mathbb{R}^{B \times K}$, which represents the after-exponential latent representations of batches of samples, and $n$ number of iterations; return the sinkhorn prototype output $\boldsymbol{Q} \in \mathbb{R}^{B \times K}$.

---

1: $\boldsymbol{Q} \leftarrow \boldsymbol{Z}^T$
2: $\boldsymbol{Q} \leftarrow \sum_i \sum_j \boldsymbol{Q}_{i,j}$
3: $\boldsymbol{U}, \boldsymbol{R}, \boldsymbol{C} \leftarrow \text{zeros}(K), \text{ones}(K)/K, \text{ones}(B)/B$
4: **for** i in 0...n **do**
5:    $\boldsymbol{U} \leftarrow \sum_j \boldsymbol{Q}_{:,j}$
6:    $\boldsymbol{Q} \leftarrow \boldsymbol{Q} * (\boldsymbol{R}/\boldsymbol{U})$
7:    $\boldsymbol{Q} \leftarrow \boldsymbol{Q} * (\boldsymbol{C}/(\sum_i \boldsymbol{Q}_{i,:}))$
8: **end for**
9: $\boldsymbol{Q} \leftarrow (\boldsymbol{Q}/(\sum_i \boldsymbol{Q}_{i,:}))^T$
10: **return** $\boldsymbol{Q}$

---

### A.1.2 PSEUDO-PARALLEL DATA MINING WITH FILTER SUITE

In addition to explanation given in the main paper, we further clarify certain note-worthy details of the Filter Suite. Firstly, we sample sentences from two monolingual corpora $s$ and $t$, and compute their cluster assignments using our LAgSwAV model with $\text{softmax}$ operation instead of sinkhorn because it is unreliable to conduct batch-wise operation during inference. Specifically, we compute the mining sentence embeddings $\boldsymbol{e}_X$ of a given sentence $X$ of any language as:

$$\boldsymbol{e}_X = \text{softmax}(\boldsymbol{C}^T \mathcal{F}_\theta(X)) \tag{11}$$

$\boldsymbol{e}_X$ is then used by the margin-based algorithm proposed by Artetxe & Schwenk (2019b) to mine pseudo-parallel data. Secondly, as the proposed Filter Suite consists of a set of different filtering criteria,, we specify the motivations for each of them as follow:

1. Minimum and maximum sentence lengths of $L_{\min}$ and $L_{\max}$: short sentences may not be useful, while long sentences may be more likely to mis-aligned.

2. Source-target length ratio $\mu$: source and target sentences should be of comparable length, the variation may depend on the specific language pair.

3. Subword overlap ratio $\gamma_i$ between source and target: for relatively close language pairs (En, Fr, De, Ro), parallel sentence pairs of the same meanings are likely to share a certain amount of subword; so we accept aligned pairs whose $\gamma_i$ is higher than the corpus average. In addition, we also cap this value up to a maximum, such as 0.35, to avoid the exact match scenario, where source and target are exactly the same. Note that we do not use this criterion for low-resource languages, such as Nepali and Sinhala.

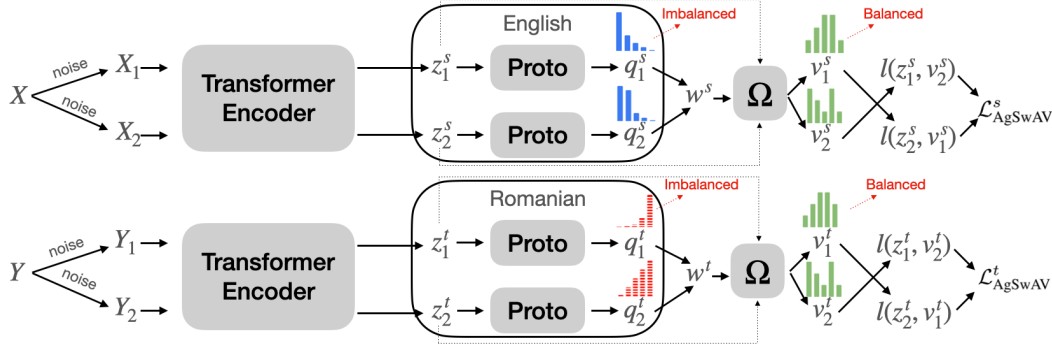

Figure 5: Complete diagram for our LAgSwAV. The diagram shows the paths each English $(X)$ and Romanian $(Y)$ sentences take to compute the final LAgSwAV loss.

4. Accept only top $\rho\%$ of pairs: sentence pairs are ranked based on alignment scores, as a result, pairs with low scores are ranked at the bottom and are often low-quality. Thus, we filter them out.

5. UMT filtering: once the UMT model has been trained to be competent enough, it can act as a filterer too. Therefore, we use it to translate the mined data to verify whether the source and target are sufficiently aligned.

### A.1.3 RANK-BASED CROSS-ENTROPY FOR UMT INTEGRATION

In addition to the total loss $\mathcal{L}$ formulation for the rank-based cross-entropy loss and the UMT integration strategy in the main paper, we further provide the overall UMT integration of the mined dataset procedures in Algorithm 2.

---

**Algorithm 2** Augmented Iterative Back-translation: Given monolingual data $\mathbb{X}_s$ of languages $s$, pseudo-parallel data $\mathbb{X}_{s-t}$, pre-trained model $\hat{\theta}$. Target side $t$ is **omitted** for clarity.

---

1: $\theta \leftarrow \hat{\theta}$
2: **while** until convergence **do**
3:     Sample batches $X_s$ from $\mathbb{X}_s$
4:     Sample batches $(X, Y)$ from $\mathbb{X}_{s-t}$, where $X = [x_1, ..., x_n], Y = [y_1, ..., y_n]$
5:     $\mathcal{L}_{BT} = -\log P_\theta(X_s | \text{translate}_\theta(X_s))$     {Standard BT loss}
6:     $\mathcal{L}_{AUG} = \sum_i -\sigma_i \log P_\theta(y_i | x_i)$     {Augmentation loss}
7:     $\mathcal{L} = \mathcal{L}_{BT} + \lambda_{aug}\mathcal{L}_{AUG}$
8:     $\theta \leftarrow \theta - \eta\nabla_\theta \mathcal{L}_\theta$
9: **end while**
10: **return** $\theta$

---

### A.2 COMPLETE LAGSWAV MODEL

Figure 5 illustrates the complete structure of our LAgSwAV model. Specifically, on the left side, we have our input data, which are English $(X)$ and Romanian $(Y)$ sentences. These inputs are perturbed into four different augmentations $(X_1, X_2, Y_1$ and $Y_2)$. These augmentations are then passed to the Transformer encoder and then the prototype layer, as well as the rebalancing block $\Omega$ to compute the final LAgSwAV loss.

### A.3 EXPERIMENTAL DETAILS

**WMT bilingual unsupervised tasks.** For the WMT'14 English-French (En-Fr), WMT'16 English-German (En-De) and WMT'16 English-Romanian (En-Ro) bilingual unsupervised translation tasks, we follow established predecessors (Lample et al., 2018c; Conneau & Lample, 2019; Song et al., 2019; Nguyen et al., 2021) to use the all the monolingual data from 2007-2017 WMT News Crawl datasets of the two languages for each task, which include 190M, 78M, 309M and 3M sentences

for English (En), French (Fr), German (De) and Romanian (Ro) respectively. We fine-tune the pre-trained XLM (Conneau & Lample, 2019) provided by the authors with our proposed LAgSwAV loss. For each language pair, we also use the respective provided bilingual dictionary of 60K sub-word units (Sennrich et al., 2016b). Regarding the mining process, we sample up to 20M sentences from the monolingual data explained above for each language and use our LAgSwAV model to mine pseudo-parallel data with Filter Suite explained in §3.2. We set mininum, maximum lengths of $L_{\min} = 5$ and $L_{\max} = 300$; source/target length ratio $\mu$ 1.5; maximum overlap ratio $\gamma_i = 0.35$ and accept only the top $\rho = 5\%$ of mined pairs. The agreement BLEU threshold is $\beta = 30$ We use the mined data to augment the MASS (Song et al., 2019) UMT models. Following all established previous work, we evaluate the models using tokenized BLEU with Moses *multi-bleu.perl* script (Koehn et al., 2007). In addition, we also report corresponding sacrebleu scores (Post, 2018) in the subscript of Table 1 for our method as well as baselines that we can reproduce.

In addition, we also use bootstrap sampling (Koehn, 2004) to test for statistical significance for the WMT bilingual unsupervised tasks. As a result, we round that our method's improvements are significant with p-value $< 0.05$ for En-Fr, En-De, and En-Ro (both directions). Plus, we also ran our experiments 3 times and found the standard deviations are approximately 0.1, 0.2, 0.1, 0.3, 0.2, and 0.3 BLEU for En-Fr, Fr-En, En-De, De-En, En-Ro and Ro-En, respectively.

**FLoRes low-resource tasks.** For the FLoRes low-resource unsupervised translation tasks (Guzmán et al., 2019), we follow the same setup as mBART (Liu et al., 2020). Specifically, we reuse their published pre-trained mBART model to fine-tune LAgSwAV loss as well as initial model for UMT back-translation training. We also use the same sentencepiece tokenizer model (Kudo & Richardson, 2018) as mBART. We sample up to 20M sentences of the English, Nepali and Sinhala corpora of the CC25 pre-training data to mine pseudo-parallel data and train the UMT model. To avoid high-rank mis-alignments in the mined data, we filter the mining monolingual data to include only sentences with unicode $\backslash p\{InDevanagari\}$ and $\backslash p\{InSinhala\}$ word counts more than 80% the sentence length for Nepali and Sinhala respectively.[2]

**T-SNE Visualization.** We sample 50K sentences of both languages from a held-out monolingual data for each language pair and use T-SNE (Van der Maaten & Hinton, 2008) to visualize the sentence embeddings of those sentences produced by the baseline XLM (encoder output of '[eos]' token) and the SwAV as well as LAgSwAV fine-tuned XLM models (softmax representations of prototype outputs $\mathrm{softmax}(\boldsymbol{C}^T\boldsymbol{Z})$). These visualizations are shown in Figure 6. As it can be seen, the sentence embeddings of English sentences (red) are scattered separately and distinguishable from those of French, German and Romanian (cyan) in XLM and SwAV XLM. Among these, the visualizations for German fare better than French's and Romanian's. However, after fine-tuning with LAgSwAV loss, the sentence embeddings are mixed indistinguishably, indicating a high degree of language-agnostic property.

**Tatoeba Accuracy.** Artetxe & Schwenk (2019a) proposed to use parts of the Tatoeba corpus as test sets for the Tatoeba similarity search task, which can be used to measure how good a model is at mining parallel data. Specifically, for each source sentence, we use cosine similarity to search for the closest target sentence based on the sentence embeddings to check if the obtained target sentence is indeed the correct target of the given source, from which we compute the Tatoeba accuracy score. In other words, given a parallel test set $[(X_1, Y_1), ..., (X_n, ..., Y_n)]$ and a nearest neighbor search function $\mathcal{N}$ operating on the sentence embeddings of the sentences, we compute the Tatoeba accuracy as:

$$Acc_{\text{Tatoeba}} = \frac{1}{2n}(\sum_i \mathbf{1}[\mathcal{N}(X_i, [Y_1, ...]) = Y_i] + \sum_i \mathbf{1}[\mathcal{N}(Y_i, [X_1, ...]) = X_i]) \qquad (12)$$

**Global Accuracy.** This metric also uses a valid parallel set to measure how lang-agnostic the models are, based on the perception that non-agnostic models exhibit a separation of latent spaces for different languages, like in Figure 3. Specifically, if there is a separation by language, each sentence of language **s** would be closer to all other sentence of the same language **s** than any sentence of the other language **t**. Therefore, we measure Global Accuracy in the same way as Alignment Accuracy, except that we conduct search for the target sentence among a search space of not just the target side, but both source and target side combined. If the model is non-agnostic, the search will always

---

[2] $\backslash p\{InDevanagari\}$ and $\backslash p\{InSinhala\}$ are unicode blocks dedicated for characters in Nepali and Sinhala.

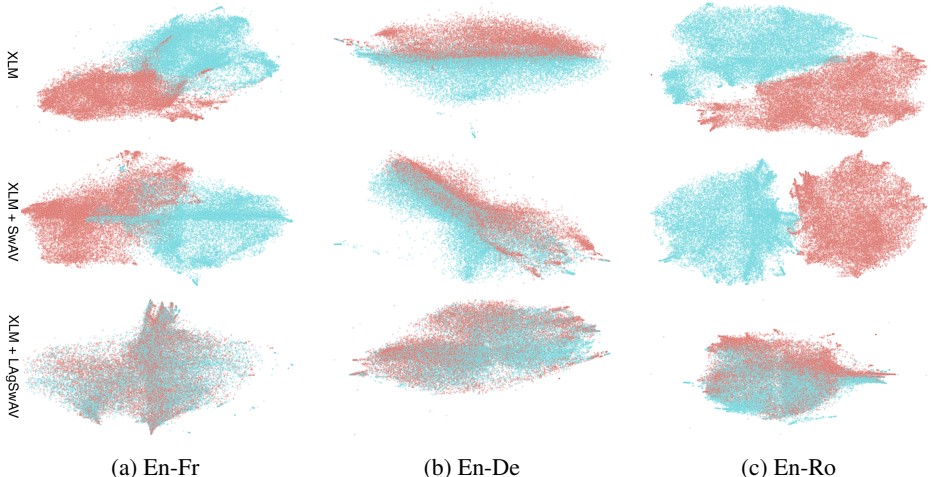

(a) En-Fr  (b) En-De  (c) En-Ro

Figure 6: T-SNE visualizations of XLM, XLM+SwAV and XLM+LAgSwAV for En-Fr, En-De and En-Ro. For each chart, we sample 50K sentences from the respective dataset and compute T-SNE on the embeddings produced by the models.

pick incorrect intra-lingual sentences, driving down the accuracy significantly. Formally, with the same notations as above, Global Accuracy is computed as:

$$Acc_{\text{glob}} = \frac{1}{2n}(\sum_i \mathbf{1}[\mathcal{N}(X_i, [X_1, ..., Y_1, ...]) = Y_i] + \sum_i \mathbf{1}[\mathcal{N}(Y_i, [X_1, ..., Y_1, ...]) = X_i]) \quad (13)$$

**Word Distribution Score.** This metric uses a held-out monolingual data to measure how semantically classifiable the models are in terms of content and topics. One of the ways to measure this is to detect certain words that appear frequently in only one cluster and rarely in others. The intuition is that if a cluster contains certain key-words dominantly, it is likely the sentences in such cluster are of similar semantic topic or content. Specifically, given same data sampled in §4.3.1, we use the K-means clustering (Vassilvitskii & Arthur, 2006) to cluster uhthe sentences into $N = 20$ clusters. Then, for each unique word $w_i \in V$ in the data, we count its appearance frequency $F_{w_i}^j$ for each cluster $j$. After divide it by the total occurance of the word $w_i$ in all clusters and compute the normalized mean scores for each word. Formally, the Word Distribution Score is computed as:

$$WoDis = \frac{1}{N}\sum_j \frac{1}{V}\sum_{w_i \in V}(F_{w_i}^j / \sum_j F_{w_i}^j) \quad (14)$$

**Percentage $\rho\%$ and Translationese effect.**

In addition to analyses of percentage $\rho\%$ for En-Ro and translationese effect for En-De, we also report the similar analyses for Ro-En and De-En, respectively. The results are shown in Figure 7, we observe that the performance of our method for unsupervised Ro-En task reduces gradually with the percentage $\rho\%$. As shown similarly in Table 7, our method is also not affected by the translationese effect for De-En direction.

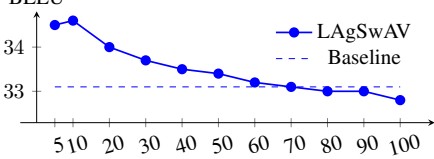

| De-En | Baseline | LAgSwAV |
|---|---|---|
| $X \rightarrow Y^*$ | 24.7 | 25.2 |
| $X^* \rightarrow Y$ | 34.2 | 38.5 |
| $X^{**} \rightarrow Y^*$ | 32.0 | 32.6 |

Figure 7: Performances of our method on Ro-En w.r.t filtering $\rho\%$.

Table 7: Translationese analysis on De-En unsupervised MT models

### A.3.1 CLUSTER SAMPLES

Semantic contrastiveness reflects the model's ability to classify sentences of similar topics and contents into the same cluster group. To demonstrate this capability of our LAgSwAV model, we randomly sample monolingual sentences from the NewsCrawl English-Romanian corpora and compute their cluster assignments using our LAgSwAV model. Then, we use a simple K-means clustering method (Vassilvitskii & Arthur, 2006) to cluster the sentences into 20 clusters. After that, we extract randomly some sentences from these clusters and report them in Table 8. As it can be seen in the **next page**, the sentences in each cluster are closely related semantically. Specifically, sentences of cluster 1 talk about *football*, while those of cluster 5 talk about *airlines* and *aviation*.

Table 8: Sample English sentences grouped by a K-means clustering method applying on their cluster assignments produced by our LAgSwAV model.

| Cluster | Sentences |
|---|---|
| 1 | - Mallorca 's Fernando Navarro and Liverpool defender Alvaro Arbeloa are also included for the first time , but there is no place in the squad for veteran Real Madrid forward Raul .
- The former Bolton boss explained that a separate groin injury would already rule Owen out for three games , meaning he would miss England 's two upcoming European qualifying games .
- David Nugent was the only substitute to score during his entire tenure with England .
- " It 's a little bit more acceptable if you come back from a goal down away from home in a derby game and get a draw .
- " I always seem to play well at Loch Lomond , but it might be time to try something different , " he said .
- This article : http : / / thescotsman.scotsman.com / features.cfm ? id = 1692282007
- " I 've played together with Cristiane for one and a half years in club football , so of course I know that they have good individual players .
- Some have accused him of acting in desperation after two lacklustre performances against Namibia and Georgia - victories by 32-17 and 14-10 respectively - put the pressure on Ireland .
- The Edinburgh outfit missed the opportunity to go top of the Clydesdale Bank Premier League last weekend when Motherwell brought their impressive nine-game sequence without loss to an end .
- United have been the most consistent team in this year 's competition and , with impressive squad depth , look better-placed to win European football 's top prize than at any time since their previous success in 1999 – yet it is unlikely they will do so without Ronaldo reproducing his Premier League form in continental competition . |
| 2 | - " The flood situation has worsened , " Assam state 's relief minister Bhumidhar Barman said , adding that thousands of villages had been inundated and 20 of 27 state districts had been affected .
- Taiwan stopped the military parades 16 years ago as it sought to improve relations with communist-ruled China following the island 's transformation to democracy .
- Relatives of the 2,000-member 13th MEU , most of whom have known for more than a month that the unit was coming home , are collectively a bit confused by the inclusion of the 13th MEU in the announcement of troop cuts , and some are even angry .
- After President Bush called General Musharraf earlier in the week , the Pakistani leader announced that national elections would be held by Feb. 15 , a month later than planned .
- CAIRO , Egypt ( AP ) – Osama bin Laden warned Iraq 's Sunni Arabs against fighting al-Qaida and vowed to expand the terror group 's holy war to Israel in a new audiotape Saturday , threatening " blood for blood , destruction for destruction . "
- Palestinian leaders are increasingly promoting the idea of a swap , provided they get comparable land in Israel , even though it implies recognition that large Jewish settlements will remain in place .
- EDT ) in Tinsukia town , east of the state 's main city Guwahati , when a bomb near a Hindu temple killed three people , officer Bhaskar Jyoti Mahanta said .
- Tuesday and was carried out by a U.S.-controlled Iraqi army unit at the Rustamiyah military academy .
- Syed Munawar Hasan , a leader of the Islamist Muttahida Majlis-e-Amal , or United Action Forum , said its lawmakers would resign from the National Assembly , Parliament 's lower house , but were keeping their options open until after the Supreme Court verdict on whether to resign from the four provincial assemblies .
- Reporters without Borders said Suleiman , in letters sent from prison , had complained of being handcuffed and beaten then put into an isolation cell where he received very little food or water . |
| 3 | - Kelly Osbourne made a triumphant debut on Monday night as prison matron Mama Morton .
- The best thing about the picture , which Harris also directed and co-wrote , is its respect for the strengths of a film form that dominated Hollywood 50 years ago but today is made only when a pedigreed actor like Harris ( or Kevin Costner or Tommy Lee Jones ) insists on it .
- Pianist Brian Kellock will make several appearances - with singer Sheila Jordan ( 1 November ) , in his BK3 collaboration with New York 's Chris Lightcap and Matt Wilson ( 25 October ) and - bringing us neatly back to Denmark - with guitar maestro Jacob Fischer ( 28 November ) .
- NEW YORK ( AP ) - Drag queens , carnival kings , Big Bird and John Travolta are all in a day 's work to Jon Coles , whose company has outfitted all of them with feathers .
- Minutes before Best Comedy was announced , " 30 Rock " star Alec Baldwin - who 'd inexplicably lost Best Actor to Ricky Gervais for his short-lived HBO comedy " Extras " - ducked past me to get a cigarette outside the theater 's back door .
- ( And why the wig from the 1940s for a play set in 1974 ?
- Says the theater : It 's a genderless story with multi-generational appeal ; plus , it 's nice and dark , just in time for Halloween . $15 - $20 ; students , $10 .
- The atmosphere - and the acting , for that matter - bring to mind Vincent Price and Victorian-era full moons .
- The Massachusetts-born Goulet , who spent much of his youth in Canada , gained stardom in 1960 with " Camelot , " the Lerner and Loewe musical that starred Richard Burton as King Arthur and Julie Andrews as his Queen Guenevere .
- The popular soprano Aprile Millo was scheduled to appear in Italo Montemezzi 's " Incantesimo , " the second of two one-act operas on fortune-telling themes performed in concert at Avery Fisher Hall on Tuesday evening , but it was apparently not in the cards . |
| 4 | - But don 't wait too long to get started : The protection didn 't kick in until the women had eaten less fat for four years and counting .
- " Based on our experience , the fact he 's moving so well , so early after such a catastrophic injury means he will walk again , " said Dr. Barth Green , chairman of the department of neurological surgery at the University of Miami school of medicine .
- Salvia 's role in brain chemistry therefore needs more research , Mr Schifano says .
- Variants of DNAJA1 crank out high or low levels of Hsp 40 .
- Lead author Mario Ferruzzi of Purdue University said catechins are relatively unstable in non-acidic environments , such as the intestines , and less than 20 percent of the total remains after digestion .
- And Dr. Berelowitz said the study had been accepted by the British Journal of Cardiology and would soon be published .
- The Company 's first theranostic product is PT-401 , a " Super EPO " ( erythropoietin ) dimer protein drug for the treatment of anemia in renal dialysis patients ( with end stage renal disease ) .
- Indeed , Dr Moffitt has already used the New Zealand group to show how a violent family upbringing and different versions of another neurologically important gene interact to produce more and less violent people .
- That 's until she met Dr. Andrew Jacono , a reconstructive plastic surgeon at North Shore University Hospital .
- " We must tend to the whole patient experience including the psychological support which patients often need . |
| 5 | - LOUIS , Oct. 19 / PRNewswire-FirstCall / – The Boeing Company ( NYSE : BA ) today announced that Utah will benefit from an estimated 600 direct and indirect jobs if it is selected to build the U.S. Air Force 's new tanker aircraft fleet .
- Entrepreneur Sir Richard Branson is backing a bid to break the world record for circumnavigating the globe in a powerboat run on 100 % renewable fuel .
- To save on fuel and other costs , Maxjet selects its departure dates and times to maximize the number of passengers , company officials said .
- The Delta pilot made a nosedive and missed the plane by about 400 feet , the Federal Aviation Administration said .
- Business-class seats will cost about 15 to 20 percent more than they do on other Singapore Airlines planes on comparable routes " because of the substantial amount of extra real estate " devoted to these seats , he said .
- Along with its twin TC-2 , TC-1 was the first satellite built and operated by the Chinese National Space Administration in cooperation with the ESA .
- She was on the 78th floor , waiting for an express elevator to leave the south tower , she said , when the second plane struck .
- Details were also given of how instrumentation developed for the Beagle 2 and Rosetta missions is being turned into a cost-effective tool for diagnosing tuberculosis .
- Chang 'e One is scheduled to scan the lunar surface from Wednesday in preparation for an unmanned moon vehicle planned for 2012 and a manned landing within 15 years .
- The site in the village of Dongchang-ni appears to be designed to launch " a bigger-sized missile or satellite projectile " than rockets deployed from the North 's east coast facility . |

