# OpenReview forum: "Contrastive Clustering to Mine Pseudo Parallel Data for Unsupervised Translation"
_ICLR.cc/2022/Conference — ICLR 2022 Poster_

### Official Review · Reviewer_NzbT · 2021-11-02

**Correctness:** 3
**Technical Novelty And Significance:** 4
**Empirical Novelty And Significance:** 2
**Recommendation:** 5
**Confidence:** 4

**Main Review:**

Strengths:
- this work is well-movited
- attempt to introduce CV SwAV loss in NLP is interesting
- the analysis is very detailed and cover most questions that the reader may have

Weaknesses:
- I found the paper extremely difficult to read. Many equations that are not necessary in the paper (maybe better in appendix) with various notations that are difficult to connect with each other.
- Table 1 is wrong. Almost none of these scores are comparable (use of different post-processing, BLEU implementations, and/or data). The paper cannot be published as long as it makes claim based on this table. Easy fix: remove all the multibleu scores.
- statistical significance testing is never performed on the results. As far as we know, they may all be coincidental.
- experiments settings are unclear (hyper-parameters, dataset pre-processing, etc.). Also I could not understand what baselines are reproduced by the authors. I am confident that I cannot reproduce the results.


Questions/comments:
- introduction: the SwAV loss is mentioned the first time without any explanation of what it is. Acronym should at least be spelt out.
- introduction can be reduced significantly. I think too many details are given on the approach that are then given again latter in the paper.
- Section 3.1; "applying masked language... to images" sounds strange. I am not sure why applying a language model to image would make sense.
- I suggest to draw a complete figure illustrating all the framework
- one of the assumption of the paper is that regular UMT may not exploit the potential parallel inside the monolingual data. it would be very interesting to check this assumption (for instance by finding and removing the parallel data from the monolingual data and retrain UMT). I actually think your assumption is wrong: UMT do exploit these parallel data and it is one of the explanation behind the very high BLEU scores obtained for en-fr.
- I found "LAgSwAV" very difficult to read. Maybe it is just the case for me, but I would recommend to lowercase more letters or just change the name to something more readable.

**Summary Of The Paper:**

This paper presents improvements of unsupervised machine translation (UMT).
It introduces a modified SwAV loss to cluster sentences from monolingual data that are then exploited during training of UMT.
Evaluation is performed on standard UMT tasks and an in-depth analysis is provided.

**Summary Of The Review:**

A good idea overall but with many details and choices difficult to understand. The paper would benefit from being restructured and simplified.
The evaluation is very questionable and prevent a publication of the paper.

---

### Official Review · Reviewer_3VMA · 2021-11-02

**Correctness:** 4
**Technical Novelty And Significance:** 4
**Empirical Novelty And Significance:** 2
**Recommendation:** 8
**Confidence:** 4

**Main Review:**

Strengths :
- The authors introduce a novel language agnostic approach to mine synthetic parallel data from existing monolingual corpus of different languages
- Adapting SwAV loss from computer vision into natural language processing as a fine-tuning step on top of masked language models.
- Re-balanced weighting technique to force a balanced cluster assignment independent of the language itself (in case of using separate masked language models) and thus a mixed embedding space
- Language agnostic embedding space  is clearly shown in their tsne plots, sentences with similar content and from different languages are close to each other in the embedding space
- The proposed approach doesn’t require pretraining on massive multilingual parallel data

Weaknesses:
- The author proposed a set of filtering criteria to filter out low quality samples from the initial-mined dataset. The filtering criteria rely more on hand-crafted thresholds like source-target length ration, and sub-word overlap which would require per language pair tuning to get a decent result. A meta-classifier on top of these features can be trained using a small clean source-target dataset to avoid per-language tuning.
- gradients are not flowing through re-balanced weights vector and the weights are computed based on per batch statistics, wouldn't it be better to optimize the re-balanced weights to avoid local bias within an already imbalanced batch (what if that batch contains sentences from source only), it would be a good idea to do an ablation study on an imbalance set of batches to see the cluster assignment behavior and its effect on the data mining.
- The target of the paper is to mine parallel data for machine translation, it makes sense to evaluate training an NMT network from scratch on mined parallel data and compare it to baselines[1, 2] mined data trained also from scratch.
- In background section pseudo-parallel data mining for machine translation, the authors missed citing some related papers that generated synthetic data for neural machine translation for example:  [4, 5]

References:

[1] Wu, Jiawei, Xin Wang, and William Yang Wang. "Extract and edit: An alternative to back-translation for unsupervised neural machine translation." arXiv preprint arXiv:1904.02331 (2019).

[2] Wu, Lijun, et al. "Machine translation with weakly paired documents." Proceedings of the 2019 Conference on Empirical Methods in Natural Language Processing and the 9th International Joint Conference on Natural Language Processing (EMNLP-IJCNLP). 2019.

[3] Song, Kaitao, et al. "Mass: Masked sequence to sequence pre-training for language generation." arXiv preprint arXiv:1905.02450 (2019).

[4] Sennrich, Rico, Barry Haddow, and Alexandra Birch. "Improving neural machine translation models with monolingual data." arXiv preprint arXiv:1511.06709 (2015).

[5] Chinea-Rios, Mara, Alvaro Peris, and Francisco Casacuberta. "Adapting neural machine translation with parallel synthetic data." Proceedings of the Second Conference on Machine Translation. 2017.

**Summary Of The Paper:**

The authors propose a language agnostic SwAV loss function to mine pseudo parallel data from unlabeled monolingual corpus from various languages. They start with a pretrained masked language model and then fine-tune it using their proposed language agnostic SwAV. The self supervised model predicts cluster code based on representation in latent space created by swapping predicted codes between different views of the same sentence. Furthermore, they extend that proposed loss with a weighting term to re-balance predictions coming from different embedding spaces resulting in a homogeneous space between source and target languages. They experiment on both Wmt14 dataset and low resource FLoRes dataset achieving state of the art bleu score compared to existing unsupervised machine translation.

**Summary Of The Review:**

Overall, the authors’ proposed approach is novel and their empirical results show the effectiveness of their approach on several datasets. I accept the paper in its current form, but with some weaknesses including the requirement to hand tune thresholds for the filter suite used on the mined data, and the paper is missing the rationale behind using per batch statistics to re-balance cluster assignment without flowing gradients through them.

---

### Official Review · Reviewer_QiHo · 2021-11-03

**Correctness:** 3
**Technical Novelty And Significance:** 3
**Empirical Novelty And Significance:** 3
**Recommendation:** 8
**Confidence:** 4

**Main Review:**

The paper is very clearly written in that I have no problem reading it and understanding the concepts. The experiments are well structured and provide reasonable depth in evaluation. The authors are also clear on the motivations, which is a good thing since it highlights the real contributions of this paper, which is enhancing the multi-lingual encoder's property of being language-agnostic, while making it more robust to noise.

A minor point that concerns me is the fact that it seems semantic contrastive. I can understand that artificial noise might not completely alter the semantic representation of the sentence to the extent that it's general context/topic becomes completely different, but I am a bit conservative when it comes to whether such improvement in robustness could really be interpreted as increase in semantic contrast. Topic and word distribution alone isn't enough to model sentiment, let alone semantics. This is a minor point in the use of terminology there, but I do wish the authors would clarify their point-of-view on the subject.

**Summary Of The Paper:**

The paper proposes a fine-tuning method for multi-lingual sentence representations trained on monolingual data, such that the final representation is more language agnostic, as well as more contrastive in the sense that artificially infused noise do not break the clustering of these representations.
The authors then experiments on using these fine-tuned sentence encoders to mine parallel samples from monolingual data, which is then integrated into UMT training, and show visible improvements.

**Summary Of The Review:**

This paper proposes empirically useful methods to fine-tune multi-lingual sentence encoder for mining data for Unsupervised NMT augmentation. I believe that this approach shows novelty, and the paper itself good motivation and inspiration from previous work, making it a pleasant read. I have only minor concerns regarding terminologies and interpretations.

---

### Official Review · Reviewer_ARKs · 2021-11-03

**Correctness:** 3
**Technical Novelty And Significance:** 3
**Empirical Novelty And Significance:** 3
**Recommendation:** 6
**Confidence:** 4

**Main Review:**

Strengths:

The idea of rebalancing cluster assignment between language makes sense and the improvement in Table 3 is very strong, showing why the modification to the loss is important. The improvement in Table 4 is also very interesting, showing that rank-based entropy and filter suite are important to make the mining effective. Overall I found the work admirable: It is novel and results are strong

Weakness:

The method itself is fairly complicate. First, SwAV itself is already complicated, its root is from computer vision, where SwAV is popular I guess because it not straightforward to do masking in computer vision. Bringing SwAV into NLP is a very nice attempt, yet I am concerning whether there is a simpler method to learn meaningful multilingual sentence embedding in an unsupervised manner so that we can mine parallel sentences. Second, extra gradients are needed, including tweaking different hyper-parameters from filter-suite and rank-based cross entropy integration (lambda hyper-parameter). This not only makes the framework complicate to scale, but also makes it hard to compare with other approaches (because we need to tune many things for the method itself but also other methods properly).

Question: Table 4 is too good to be true because LAgSwAV is very close to LASER, a supervised method. I guess I am not satisfied with this and more explanation why it is the case could help.






**Summary Of The Paper:**

This work provides a nice extension to the SwAV proposed by Caron. Specifically, it proposes a variant that can be used to build meaningful cluster assignment for different languages at the same time in an unsupervised manner. The key to their modification to SwAv is the so called language-agnostic rebalancing function, which is a simple modification that encourages the cluster assignments to sentence in different languages balance. Experiments show that their modification improves mining quality (i.e. resulting higher BLEU score) than using SwAV. Despite that, the work shows that using their model is not enough, and we need extra steps to make the mined data usuable (i.e. using filter suite to filter data and rank-based cross entropy to train the model with a better use of the data).

**Summary Of The Review:**

An admirable work on mining parallel sentences from building meaningful sentence cluster assignment. Tweaks including rank-based entropy and filter suite are important to make it work.

---

### Author Response · Authors · 2021-11-22
**New Revision**

Dear Reviewers,

We have uploaded a new revision to our paper, in which we incorporate some of the comments from the reviewers. Specifically, we mainly:
1. Added a complete model diagram of our LAgSwAV model in the Appendix (page 14) to facilitate better method understanding.
2. Added statistical significance analysis in the Appendix (page 15)
3. Added suggested citations.

We hope the reviewers have another look at the paper.

As we will no longer be able to reply to the reviewers' questions after the deadline, we sincerely hope the reviewers acknowledge and respond to our responses to your reviews, if you have not done so. Your feedback is extremely valuable to us to improve our work!

We deeply appreciate your time and effort in reviewing our paper.

Best regards,

---

### Decision · Program_Chairs · 2022-01-20

**Decision:**

Accept (Poster)

**Comment:**

Overall, reviewers are positive. The majority praised the approach as novel and viewed the results as quite strong. Further, reviewers valued the provided ablations and analysis that helped motivate the proposed method. A few concerns were raised about overall clarity (though some reviewers praised the clarity of presentation), the use of hand-crafted filters, and certain experimental comparisons. The majority of these concerns have been adequately addressed in author response.